# Potential Effect of Polyphenolic-Rich Fractions of Corn Silk on Protecting Endothelial Cells against High Glucose Damage Using In Vitro and In Vivo Approaches

**DOI:** 10.3390/molecules26123665

**Published:** 2021-06-16

**Authors:** Nurraihana Hamzah, Sabreena Safuan, Wan Rosli Wan Ishak

**Affiliations:** 1Nutrition & Dietetics Department, School of Health Sciences, Universiti Sains Malaysia, Kubang Kerian 16150, Kelantan, Malaysia; nurraihanahamzah@gmail.com; 2Biomedicine Department, School of Health Sciences, Universiti Sains Malaysia, Kubang Kerian 16150, Kelantan, Malaysia

**Keywords:** corn silk, polyphenolic-rich fraction, endothelial cell, HUVECs, anti-diabetic activity

## Abstract

Endothelial cell dysfunction is considered to be one of the major causes of vascular complications in diabetes. Polyphenols are known as potent antioxidants that can contribute to the prevention of diabetes. Corn silk has been reported to contain polyphenols and has been used in folk medicine in China for the treatment of diabetes. The present study aims to investigate the potential protective role of the phenolic-rich fraction of corn silk (PRF) against injuries to vascular endothelial cells under high glucose conditions in vitro and in vivo. The protective effect of PRF from high glucose toxicity was investigated using human umbilical vein endothelial cells (HUVECs). The protective effect of PRF was subsequently evaluated by using in vivo methods in streptozotocin (STZ)-induced diabetic rats. Results showed that the PRF significantly reduced the cytotoxicity of glucose by restoring cell viability in a dose-dependent manner. PRF was also able to prevent the histological changes in the aorta of STZ-induced diabetic rats. Results suggested that PRF might have a beneficial effect on diabetic patients and may help to prevent the development and progression of diabetic complications such as diabetic nephropathy and atherosclerosis.

## 1. Introduction

The vascular endothelium is a single layer of cells that covers the inner surface of all blood vessels. In addition to forming a physical barrier to protect the vessel wall, the endothelium secretes a large number of bioactive substances involved in the modulation of vascular tone, coagulation, cell proliferation, and inflammation. However, endothelial cells are unable to regulate glucose transport as well as other cells do. Therefore, they are more vulnerable to the toxic effects of hyperglycemia [1]. Hyperglycemia can affect endothelial and other vascular cells at the cellular level, delay endothelial cell replication and cause excessive cell death. Hyperglycemia-induced reactive oxygen species (ROS) production induces RNA and DNA damage that may be responsible for the reduced proliferation rate observed in endothelial cells. Studies have indicated that ROS can induce apoptosis in endothelial cells. The involvement of ROS in the induction of apoptosis by high glucose has been demonstrated in human umbilical vein endothelial cells (HUVECs) [2,3].

Oxidative stress occurred when the ROS level in the human body increased due to prolonged hyperglycaemia [4]. Oxidative stress can cause functional disorders of the endothelial layer and thus, in turn, cause damage to the function and structure of vascular tissues, such as the aorta [5]. Furthermore, damage to vascular endothelium is the primary complication of type 2 diabetes mellitus leading to endothelial dysfunction and further complications such as diabetic nephropathy [6]. This suggests the requirement of effective antioxidant therapy for neutralizing the harmful effects of ROS on vascular cells. Recently, the use of bioactive components has been considered as a new approach in the prevention and management of diabetes and its complications [7]. Plant-derived phenolic compounds possess a wide range of pharmacological properties and their mechanisms of action have become the subject of considerable interest in recent years. In this context, a phenolic-rich fraction received much attention because of its potent free radical scavenging and antioxidant action [8].

Wastes of fruits and vegetables have been reported to be rich in phenolic compounds, which are considered to have a beneficial effect on health. Corn silk is a waste by-product of corn cultivation. Corn silk is predominantly discarded together with other parts of the plant due to a lack of effective utilization. However, corn silk has been reported to exhibit anti-diabetic, antioxidant, diuretic kaliuretic, and anti-depressant effects [9]. Moreover, corn silk has been reported to contain a high level of phenolic compounds that possess antioxidant activity [10]. In this aspect, the PRF of corn silk (PRF) can be tapped as a potential natural antioxidant that can be used to prevent oxidative stress. Therefore, the present study investigated the potential protective role of PRF against injuries to vascular endothelial cells under high glucose conditions in vitro and in vivo.

## 2. Results

### 2.1. Protective Effect of PRF against High Glucose in Cytotoxicity in HUVECs

Figure 1 shows the effect of PRF and metformin on cell viability in HUVECs treated with high glucose of 30 mM. When HUVECs were treated with 30 mM glucose for 24 h, the cell viability significantly decreased (*p* < 0.05) by 25.09% compared to HUVECs treated with low glucose of 5.5 mM. However, metformin and PRF protected the cells from high glucose-induced damage by significantly increasing the cell viability (*p* < 0.05).

Table 1 shows that lower concentrations of PRF compared to metformin are required to enhance the proliferation of HUVECs, indicating that PRF was more effective than metformin in protecting HUVECs from a high glucose environment.

### 2.2. Effect of PRF on the Aorta of STZ-Induced Diabetic Rats

STZ-induced diabetic rats were treated with PRF as described in the Methods section. The morphological changes of the aorta in the PRF-treated animals in comparison to the normal control group are shown in Table 2. The aortic section of the normal control group revealed the normal thickness of the aorta wall. Tunica media thickness was measured as 138.89 ± 3.94 μm in this group (Figure 2).

A continuous layer of endothelial cells in the tunica intima can be observed in the normal control group (Figure 3A). This group also showed a parallel muscle of tunica media apparently lacking foam and fat cells. The aortic section of the diabetic control group showed moderate deposition of foamy cells in the tunica media layer (Figure 3B). Discontinuous endothelial cells at the tunica intima were observed in the diabetic control group. The mean thickness of the tunica media decreased significantly (*p* < 0.05) in the diabetic control group (172.03 ± 12.86 µm) compared with the normal control group.

The aortic section of the low dose PRF-treated group (100 mg/kg) demonstrated mild deposition of foamy cells in the tunica media and discontinuous endothelial cells of the tunica intima (Figure 3C). The mean thickness of tunica media was 152.28 ± 4.54 µm in the low-dose PRF-treated group. The aortic section of the high dose PRF-treated group (200 mg/kg) showed mild deposition of foamy cells in the tunica media (Figure 3D). However, the endothelial cells at the tunica intima were more intact compared to the low-dose PRF-treated group and metformin-treated group. The mean thickness of the tunica media was 141.18 ± 6.04 µm in the high-dose PRF-treated group. The aortic section of the metformin-treated group showed restoration of the structure back to nearly normal (Figure 3E). However, mild deposition of foamy cells was observed in the tunica media as well as discontinuous endothelial cells at the tunica intima. The mean thickness of the tunica media was 153.31 ± 6.59 µm in the metformin-treated group.

## 3. Discussion

An in vitro model of diabetes was studied to evaluate the potential of the PRF on protecting HUVECs as an endothelial cell model from high glucose-induced cytotoxicity before proceeding further with an in vivo study. Previous studies have reported that high concentrations of glucose can cause endothelial cell damage, leading to increased cell apoptosis in diabetes [2,11,12]. In this study, PRF treatment significantly improved endothelial cell viability from a high-glucose environment in a dose-dependent manner. Therefore, this PRF is capable of protecting the endothelial cells against glucose toxicity in the in vitro model. The tested PRF can increase the viability of high glucose-induced HUVECs.

To the best of our knowledge, there was no previous study done on the protective effect of corn silk against glucose-induced cytotoxicity of HUVECs. Nevertheless, Min and Han [13] reported that *Polyopes lancifolia* extract protected HUVECs from high glucose-induced damage. Another study has reported that baechu kimchi added Ecklonia cava extract protected HUVECs from high glucose-induced damage by increasing cell viability [14].

The present result shows that the PRF is more effective than metformin for the protection of HUVECs from high glucose. This protective effect might be due to phenolic compounds, which are present at a higher level in this fraction as reported in our previous study [15]. The PRF contained 22 phenolic compounds comprising of 21 flavonoids and one chalcone. The flavonoids detected in this fraction are classified into flavones, flavonols, flavone C-glycoside, flavonol O-glycosides, flavanols, and isoflavonoids [15]. Previous studies have reported that phenolic compounds, such as α-mangostin and maysin, had shown a protective effect against high glucose-induced damage in HUVECs [16,17,18,19]. Phenolic compounds can protect cells from high glucose-induced damage via inhibition of protein glycation and reduction of advanced glycation end product (AGE) formation [20,21]. Exposure of HUVEC to high concentrations of glucose-induced accumulation of AGEs. AGEs block nitric oxide activity in the endothelium and cause the production of reactive oxygen species [22]. A study by Wang and Zhao [23] showed that corn silk (CS) from mature corn fruit and its fractions had an inhibitory effect on the formation of AGEs. Therefore, PRF from baby corn used in this present study might have similar potential as the mature corn fruit to exhibit anti-glycation function via its AGE inhibitor property thus, protecting HUVECs from high glucose-induced damage.

The vascular endothelium, which regulates the passage of macromolecules and circulating cells from blood to tissues, is a major target of oxidative stress, playing a critical role in the pathophysiology of several vascular diseases and disorders [24]. Oxidative stress generation accompanies an acute increase in glucose concentration which leads to endothelial dysfunction [25,26]. Endothelial dysfunction is known to be an early step in the development of atherosclerosis, which is plaque build-up on the arterial wall. Arteries are important to humans because they are blood vessels that carry oxygen-rich blood away from the heart and to other parts of the body, and the largest artery is the aorta [27]. Therefore, this present study aimed to evaluate the effect of PRF on histological changes of the aorta in diabetic rats.

The aorta of the diabetic control group showed moderately discontinuous endothelial cells at the tunica intima and the presence of foamy cells at the tunica media. Foam cells are lipid-laden monocytes that accumulate in the vascular wall that is driven by endothelial dysfunction. These foam cells can form an atherosclerotic plaque, an early sign of atherosclerotic disease [26]. It has also been demonstrated that damage, apoptosis and death of endothelial cells have been implicated in the development of atherosclerosis and other vascular diseases [25]. Furthermore, there is growing evidence that shows that high levels of glucose impair endothelial cell replication and accelerate cell death of endothelial cells in culture. This is in line with the result of the present study showing that the viability of HUVECs decreased in high glucose media.

However, administration of PRF and metformin ameliorated the pathological changes in the aorta of diabetic rats by restoring endothelial cell layers and reducing the number of foamy cells. Besides, the aorta of the high dose PRF-treated group showed more intact endothelial cells compared to low dose PRF-treated and metformin-treated. This is in line with findings in the present in vitro study, which showed that as the concentration of PRF treatment increased, the endothelial cell viability under a high-glucose environment was increased. Taken together, PRF might ameliorate pathological changes in the aorta of diabetic rats, possibly through its antioxidant properties. According to previous studies, the antioxidants would protect the aorta from further oxidative damage [28,29]. This condition would then improve the pathological changes of the aorta, thus preventing the development of atherosclerotic lesions, as shown in the present study.

## 4. Materials and Methods

### 4.1. Sample Collection and Preparation of Fractions

Fresh baby corn (vegetable type) was purchased from a local farmer in Kampung Tendong, Pasir Mas District, Kelantan state of Malaysia. The plant’s voucher specimen (authentication number: 11801) was deposited to the Herbarium Unit, School of Biology Science, Universiti Sains Malaysia. Four batches of fresh baby corn comprising a total of 50 kg were purchased in October 2016, November 2016, April 2017 and May 2017. The fraction was prepared according to our previous study [15]. Briefly, the fresh corn silk was detached from the fruit stalks and then oven-dried (Memmert, Schwabach, Germany) at 55 °C for 48 h. The dried corn silk was ground into a powder form, using a food grinder (Tommy Tech, Shenzhen, China). The powder was sieved with a 250 µm-size sieve shaker (Retsch, Haan, Germany) and kept in an airtight Duran bottle and stored at 4 °C prior to analyses. The CS powder was extracted using a hot solvent extraction method. 5 g of CS powder was mixed with 10 mL of 40% ethanol in a 500 mL beaker. Then, the mixture was shaken using a water bath shaker (Memmert, Schwabach, Germany) for 30 min at 50 °C. The extract was centrifuged at 1700× *g* for 10 min (Hettich, Tuttlingen, Germany) and filtered through a Whatman No. 1 filter paper. The clear solution was collected and concentrated in a rotary evaporator (Eyela, Bohemia, NY, USA). The concentrated 40% ethanol extract was then sequentially fractionated using hexane (HmbG, Hamburg, Germany) and then with ethyl acetate (HmbG, Hamburg, Germany). The solvent in each fraction was removed using a rotary evaporator. Preliminary screening showed that ethyl acetate fraction showed higher phenolic content and DPPH radical-scavenging activity than the crude extract, hexane fraction and aqueous fraction [15]. Thus, the ethyl acetate fraction was selected as a phenolic-rich fraction (PRF) and used for in vivo studies.

### 4.2. Protective Effect of PRF on Primary Cell Culture

#### 4.2.1. Cell Culture

Human umbilical vein endothelial cells (HUVECs) were purchased from PromoCell. Endothelial Cell Growth Medium 2 (PromoCell, Heidelberg, Germany) supplemented with 0.02 mL/mL fetal calf serum, 5 ng/mL epidermal growth factor (recombinant human), 10 ng/mL basic fibroblast growth factor (recombinant human), 20 ng/mL insulin-like growth factor (R3 IGF-1), 0.5 ng/mL vascular endothelial growth factor 165 (recombinant human), 1 µg/mL ascorbic acid, 22.5 µg/mL heparin and 0.2 µg/mL hydrocortisone were added to the basal media. The cultures were maintained in a humidified atmosphere containing 5% CO_2_ at 37 °C until reaching 80% confluence and then passaged. Cells from passage three to nine were used for all experiments. The low passage was used to ensure low senescent changes and loss of useful replicative potential of the cells.

#### 4.2.2. Protective Effect of PRF on HUVECs from High Glucose-Induced Cytotoxicity

Cells (3 × 10^4^ cells/well), cultured in 96 well plates in a humidified atmosphere containing 5% carbon dioxide (CO_2_) at 37 °C, were pre-incubated with normal glucose (5.5 mM) and high glucose (30 mM) for 24 h, followed by treatment with various concentrations (0.001–10 mg/mL) of PRF for 24 h [10,30]. The media was carefully removed from each well, and replaced with 0.1 mL of fresh media containing 10 µL of MTT (Merck, Darmstadt, Germany). After incubation for 4 h, 100 µL of solubilization solution was added to dissolve precipitated formazans. The absorbance was measured at 570 nm by a microplate reader (Thermo Scientific, Waltham, MA, USA)). Each extract and control was assayed in duplicate in three independent experiments. Cell viability was calculated according to the following equation:The percentage of cell viability = OD of treated cells/OD of control cells × 100
where: OD = optical density.

### 4.3. Effect of PRF on the Aorta of STZ-Induced Diabetic Rats

#### 4.3.1. Dosage Preparation of PRF and Drug

Dosage calculation and stock solution preparation of PRF in this study were adapted according to the method of Erhirhie et al. [31]. The PRF was dissolved in 1% carboxymethylcellulose (Fisher, Waltham, UK) in normal saline (0.9% sodium chloride (Sigma, Saint Louis, MO, USA)) to get a stock solution of 80 mg/mL and was kept in a refrigerator at 4 °C prior to treatment. Rats were weighed individually every day, and the required dose volume of PRF for each rat at a standard dose of 100 or 200 mg/kg was calculated as follows:Calculated injected volume (mL) = [Animal weight (kg) × Dose (mg/kg)]/Concentration (mg/mL).

A similar procedure of dosage preparation was also conducted with the oral hypoglycaemic agent, metformin (150 mg/kg) (Metcheck 850, Mumbai, India) which was used as a positive control.

#### 4.3.2. Experimental Animals

Thirty male Sprague–Dawley strain rats between 2–3 months of age and weighing 250 g to 300 g were used in the present study. The rats were supplied by the Animal Research and Service Centre (ARASC), Universiti Sains Malaysia, Health Campus, Kelantan. All rats were healthy and maintained according to the institution’s ethical guidelines under standard laboratory conditions. They were housed in standard cages at an ambient temperature of 20 ± 2 °C with 12 h light or dark cycle (lights on from 0700 to 1900 h) and were placed in the animal holding room at the ARASC, USM. They were fed with commercial rat pellets (Gold Coin, Penang, Malaysia) and tap water ad libitum. They were quarantined for one week, prior to the experiment to be acclimatized. The experiment was designed and conducted according to ethical approval, cleared by the Animal Ethics Committee, USM (USM/IACUC/2017/ (832)).

#### 4.3.3. Induction of Experimental Diabetes

Prior to the induction of diabetes, rats fasted overnight. Streptozotocin (STZ) (55 mg/kg body weight) (Merck, Darmstadt, Germany) was freshly dissolved in 0.1 M cold sodium citrate buffer (pH 4.5) (R&M, Essex, UK) and was then injected once intraperitoneally (IP) to the rats. Food and water intakes were monitored daily after STZ administration. The development of diabetes in rats was confirmed by determining the fasting blood glucose level in rats’ blood taken from the tail vein at 72 h and on day seven post-STZ-injection. Glucose measurement was performed with an Accu-Chek glucometer (Roche, Mannheim, Germany). Rats showing fasting blood glucose levels ≥ 13 mmol/L were considered diabetic.

#### 4.3.4. Study Design

24 diabetic rats were randomly divided into four groups comprising of 6 rats each. each. The 5 groups used in the animal experiments were (1) Non-diabetic rats (normal control), (2) Diabetic untreated rats (diabetic control), (3) Diabetic rats treated with PRF (dose: 100 mg/kg/day), (4) Diabetic rats treated with PRF (dose: 200 mg/kg/day) and (5) Diabetic rats treated with metformin (dose: 150 mg/kg/day). Treatment of rats began on the seventh day after STZ administration, and this was considered as the first day of treatment (D1) (baseline). The rats were treated once a day for up to four weeks (D28) by depositing a solution of the compound into the esophagus using a syringe with a gavage needle (blunt-ended needle cannula). The rats were monitored for 5 min following dosing to ensure that they did not regurgitate the content or were not injured.

#### 4.3.5. Histopathological Analysis

Sections of the aorta from all groups were observed for the histopathological changes under a light microscope (Leica, Wetzlar, Germany). Photomicrographs were taken by using an image analyzer (Leica, Wetzlar, Germany). The stained sections were also evaluated by a blind histologist and the morphological changes of each slide were examined and the severities of histopathological changes in the tissue were scored as follows: none (−), mild (+), moderate (++) and severe (+++) damage.

#### 4.3.6. Measurement of the Thickness of the Aorta

The aortic sections were studied under 40× magnification using a photographic microscope and photomicrographs were captured. Aortic tunica media thickness was measured from four different points of cross-section and expressed as an average using Image J version 1.52n (National Institutes of Health (NIH), Maryland, MD, USA) [32].

### 4.4. Statistical Analysis

Data were analyzed using Microsoft Excel data analysis tool package (Microsoft, Washington, WA, USA) and IBM SPSS Statistics Data Editor Version 24 (IBM, New York, NY, USA) All data were expressed as means ± standard error of the means (S.E.M). A *p*-value of less than 0.05 was considered significant. The statistical analysis was evaluated using one-way analysis of variance (ANOVA), performed with the Post-hoc Tukey test.

## 5. Conclusions

PRF has the potential in protecting the endothelial layer from high glucose by restoring cell viability in a dose-dependent manner. PRF also was able to prevent the histological changes in the aorta of STZ-induced rats. Results suggest that such agents may have a beneficial effect on diabetic patients and may help to prevent the development and progression of diabetic complications such as diabetic nephropathy and atherosclerosis. The results of this study suggest that locally available underutilized agricultural products are of potential interest in high-value applications, namely for the food, pharmaceutical and cosmetic industries.

## Figures and Tables

**Figure 1 molecules-26-03665-f001:**
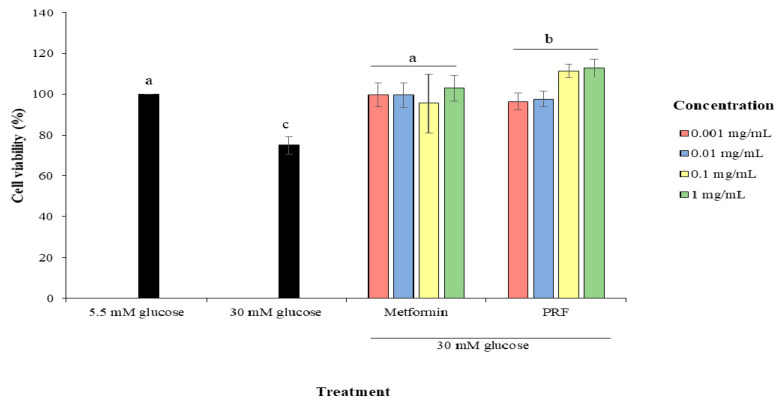
Protective effect of PRF of corn silk on the cell growth of HUVECs treated with 30 mM glucose for 24 h. PRF = Polyphenolic rich fraction. Values marked by different letters (a, b, c) are significantly different (*p* < 0.05).

**Figure 2 molecules-26-03665-f002:**
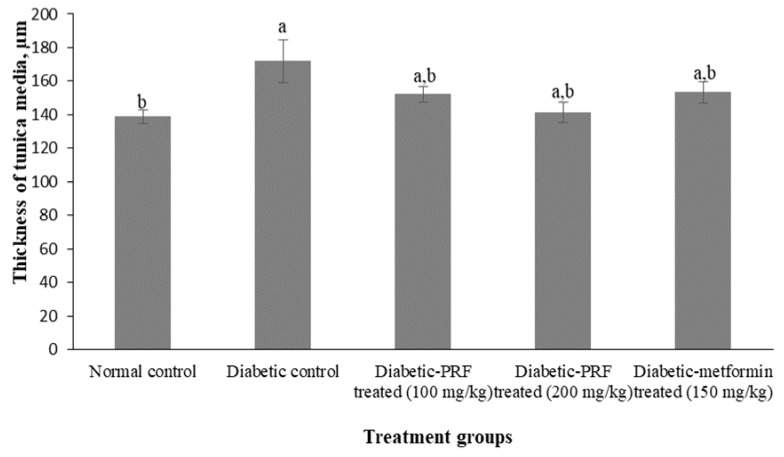
The mean thickness of tunica media for all groups. Values marked by different letters (a, b) are significantly different (*p* < 0.05).

**Figure 3 molecules-26-03665-f003:**
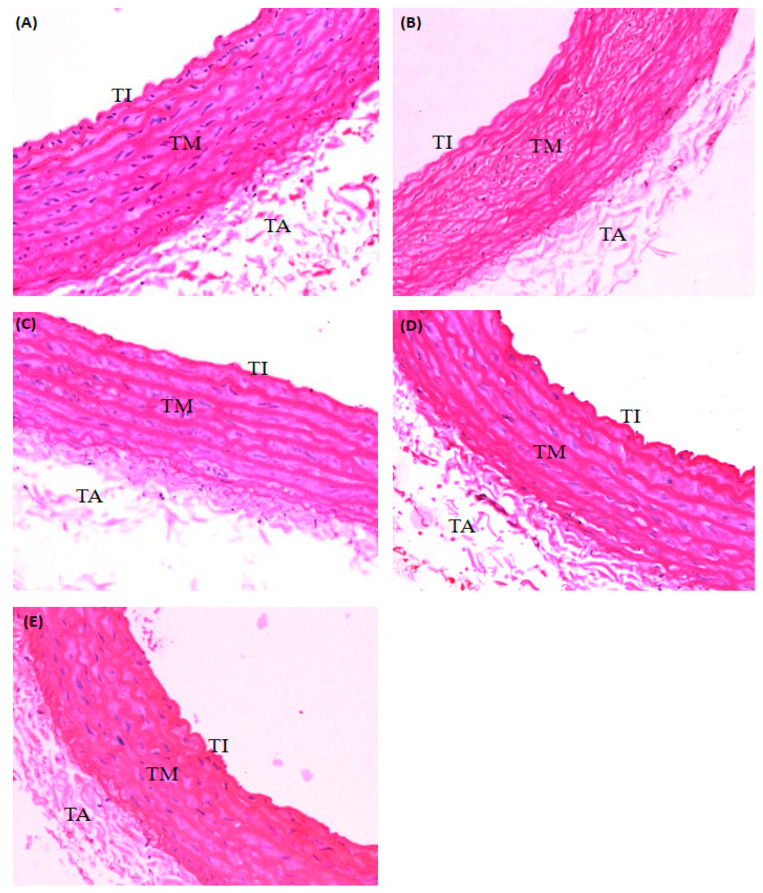
H&E staining of rat aorta captured under 40× magnifications. (**A**): Aorta of the normal control group shows the continuous layer of endothelial cells at the tunica intima; (**B**): Aorta of the diabetic control group shows an increasing thickness of tunica media compared to normal control with deposition of foamy cells and discontinuous endothelial cells at the tunica intima; (**C**): Aorta of diabetic rats treated with 100 mg/kg of PRF shows mild deposition of foamy cells in the tunica media and discontinuous endothelial cells at the tunica intima; (**D**): Aorta of diabetic rats treated with 200 mg/kg of PRF shows intact endothelial cells at the tunica intima; (**E**): Aorta of diabetic rats treated with 150 mg/kg of metformin shows mild deposition of foamy cells in the tunica media and discontinuous endothelial cells at the tunica intima. TI, tunica intima; TM, tunica media; TA, tunica adventicia. (H&E, scale bar 50 µm)

**Table 1 molecules-26-03665-t001:** EC_50_ values of PRF and metformin on HUVECs treated with 30 mM glucose.

Sample (*n* = 6)	EC_50_ (ng/mL) ± S.E.M.
PRF	108.00 ± 45.22 ^a^
Metformin	261.23 ± 112.18 ^b^

PRF = Phenolic-rich fraction of corn silk. Values marked by different letters (a, b) are significantly different (*p* < 0.05).

**Table 2 molecules-26-03665-t002:** Grading of the morphological changes in the aorta of rats.

	Histopathological Changes	Evaluation of Deposition of Foamy Cell in Tunica	Evaluation of Density of Endothelial Cells at Tunica Intima	Evaluation of Thickness of Tunica Media
Treatment Groups (*n* = 6)				
Normal control		−	−	−
Diabetic control		++	++	++
Diabetic-PRF treated (100 mg/kg)		+	+	+
Diabetic-PRF treated (200 mg/kg)		+	+	+
Diabetic-metformin treated (150 mg/kg)		+	+	+

Note: (−) none, (+) mild, (++) moderate.

## Data Availability

Data are contained within the article.

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
