# Peer review of "Potential Effect of Polyphenolic-Rich Fractions of Corn Silk on Protecting Endothelial Cells against High Glucose Damage Using In Vitro and In Vivo Approaches"

_molecules, 2021, doi:10.3390/molecules26123665_

Round 1
Reviewer 1 Report
The authors performed in vitro studies with HUVECs and in vivo studies with diabetic rats and dissected aorta that showed that a polyphenol-rich fraction from corn silk (PRF) was able to protect endothelial cells and the aorta from high glucose-induced damage and cell death.
Major comments: The study was well-designed, and the paper is well-organized. Agricultural waste is an important untapped reservoir for the extraction of polyphenols for the improvement of human health.
Figure 1: Error bars indicating SEM are required on all the bars to show that at least 3 biological replicates were performed.
The data in Table 2 is qualitative. Is there any way that any of it can be quantitated to make it more robust?
Minor comments:
There are many issues with English grammar and word usage. Recommended substitutions are shown below.
Lines 19, 22, 59, 68, 73, 81, 85, 129, 133, 144, 296: PRF of corn silk -> PRF
Line 28: Blood -> The blood
Figure 1 X-axis: Corn silk -> PRF
Figure 1 X-axis: Treatment agent -> Treatment
Line 77: are required to enhance the proliferation of HUVEC cell compared to metformin -> compared to metformin are required to enhance the proliferation of HUVEC cells
Line 79: other fractions -> metformin
Line 81: value between -> values of
Lines 81, 130, 137, 139, 142, 148, 151, 156, 177, 211, 221: HUVEC -> HUVECs
Line 86: Add an introductory sentence to this paragraph such as “STZ-induced diabetic rats were treated with PRF as described in the Methods section.
Line 86: experimental -> PRF-treated
Line 87: were -> are
Line 96: In Figure 2 please indicate if the bars marked a,b are statistically different than the bar labeled a or the bar labeled b. That appears to be the case. What is the p-value between Diabetic PRF-treated (200 mg/kg) and Normal control? The magnitude of these bars are similar.
Line 99: without apparent foam and fat cell observed -> apparently lacking foam and fat cells
Lines 101, 111, 112: at -> at the
Lines 102, 117, 118, 122, 126: of -> of the
Lines 107, 109, 114, 121, 126, 169: at -> at the
Lines 111, 114, 117, 120, 125: in -> in the
Line 120: cell -> cells
Line 124: near to -> back to nearly
Line 129: In vitro -> An in vitro
Line 131: with -> with an
Line 134: improves -> improved
Line 135: these -> this
Line 136: cell -> cells
Line 144: to protect HUVEC from high glucose compared to metformin -> than metformin for the protection of HUVECs from high glucose
Line 147: had shown -> show
Line 150: products (AGEs) -> product (AGE)
Line 153: CS -> corn silk (CS)
Line 156: anti-glycation -> anti-glycation function
Line 156: inhibitor property thus, protect -> inhibitory properties, thus protecting
Line 161: lead -> leads
Line 162: early steps -> an early step
Line 163: atherosclerosis which refer to plaque builds up -> atherosclerosis, which is plaque build up
Line 164: human -> humans
Line 165: to the heart and -> away from the heart and to
Line 166: aims -> aimed
Line 168: moderate -> moderately
Line 169: cell -> cells
Line 170: has -> is
Line 171: will -> can
Line 174: showed -> showing
Line 175: could impair -> impair
Line 176: This -> This is
Line 176: result of the present study showed -> results of the present study showing
Line 179: cells -> cell
Line 181: cell -> cells
Line 184: will be -> was
Line 185: the previous study, the anti-oxidant -> previous studies, the antioxidant
Line 188: prevent -> preventing
Line 188: lesion -> lesions
Line 191: fraction -> fractions
Line 200: gram -> grams
Line 205: acetate -> acetate.
Line 205: ethyl -> the ethyl
Line 207: fraction -> fractions
Line 212: 0.02 mL -> 2% (or 0.02 mL/mL)
Line 222: 104 -> 104
Line 223: 37oC -> 37oC
Line 227: four h -> 4 h
Line 227: were -> was
Line 253: pellet -> pellets
Line 260: rats -> the rats
Line 267: each; -> each. The 5 groups used in the animal experiments were:
Line 268: rat -> rats
Line 275: within 5 min of -> for 5 min following
Line 276: injured -> were not injured
Line 279: microscope photomicrographs -> microscope. Photomicrographs
Line 284: thickness -> the thickness
Line 285: 40× under -> 40× magnification using
Line 287: average. -> average
Line 298: suggested -> suggest
Line 299: agents -> agents may
Line 301: study would help in promoting the use of -> results of this study suggest that
Line 302: for -> are of
Line 303: in -> for the
Reviewer 2 Report
Dear Authors, The authors presented a manuscript entitled “Potential effect of polyphenolic-rich fraction of corn silk on protects endothelial cells against high glucose damage using in vitro and in vivo approaches.” Globally, the main idea of the manuscript is interesting. However, the authors must seriously explain the new findings. Some of the most important concerns and recommendations are the following: I recommend reorganizing the introduction section to have a concise explanation, for example, include details of the main phenolic / bioactive compounds found in corn silk. Other applications on the use of corn silk or scientific work. Material and methods Please, add the amount of material collected and in how many times/lots; What is the month and year of the collection? Was an antioxidant activity test performed on the extract? Please insert the characterization of the chromatographic profile of the phenolics found in the extract. Line 207: Thus, the ethyl acetate fraction was selected as a phenolic rich fraction (PRF) and used for in vivo studies. How did you remove the ethyl acetate fraction? Then it was re-suspended and what solution? Line 219: The low passage was used to ensure low senescent changes and loss of useful replicative potential of the cells. Please specify which passages were used in the experiment Line 145: This protective effect might be due to phenolic compounds, which are present at a higher level in this fraction. To make this statement, the authors must insert the results found in the antioxidant activity. References: the last reference addressed by the authors was from 2018. I advise the authors to review more current studies to support the results found.Author Response
Please see the attachment

Round 2
Reviewer 1 Report
The authors made the changes I suggested in the previous round of review. But several minor wording mistakes were introduced that are identified below. The manuscript is much improved.
Major comments: none
Minor comments:
p. 1 Introduction line 1: lood -> vascular
p. 2 line 13: diuresis -> diuretic
p. 2 line 13: and kaliuresis -> kaliuretic, and
p. 2 line 13: effect and anti-depressant -> anti-depressant effects
p. 2 line 14: high phenolic compounds and possessed -> high levels of phenolic compounds that possess
p. 2 Results line 1: on HUVECs from high glucose-induced cytotoxicity -> against high glucose-induced cytotoxicity in HUVECs
p. 3 Figure 1 legend top line: HUVECs cell -> HUVECs [the C in HUVEC already indicates cells]
p. 3 paragraph below Figure 1 legend, line 2: HUVECs cells -> HUVECs
p. 3 paragraph below Figure 1 legend, line 4: under -> from
p. 6 last line of the Results section: tunica media -> the tunica media
p. 6 Discussion line 6: under -> from
p. 6 Discussion line 12: HUVECs cells -> HUVECs
p. 6 Discussion line 24: had show -> showed
p. 6 last line: the foamy -> foamy
p. 7 line 1: Foam cells are an accumulation of lipid-laden monocytes in the vascular wall that is -> Foam cells are lipid-laden monocytes that accumulate in the vascular wall
p. 7 line 6: showing -> shows that
p. 7 Discussion line 4 from bottom of section: the antioxidant agents -> antioxidants
p. 9 Conclusions line 6: This results -> The results
Reviewer 2 Report
Dear Authors
The work has improved substantially and presents interesting results. The data would be better discussed if the analysis of antioxidant activity and the identification of phenolic compounds had been performed on the extract obtained. It is always important to carry out these analyses, as there are many variations due to numerous factors, such as soil, seasonality, incidence of sunlight, harvest, cultivation conditions, soil, storage after harvesting. Furthermore, the solvent and the extraction method also influence the quality and quantity of extracted compounds. Thus, I suggest that authors in future investigations add such methods, so as not to argue with literature data.
